# Amine-responsive cellulose-based ratiometric fluorescent materials for real-time and visual detection of shrimp and crab freshness

Ruonan Jia[1,2], Weiguo Tian[1], Haotian Bai[3], Jinming Zhang ◉ [1], Shu Wang ◉ [2,3] & Jun Zhang ◉ [1,2]

Herein, we design and prepare cellulose-based ratiometric fluorescent materials with superior amine-response, which offers the real-time and visual detection of seafood freshness. Through utilizing the reactive hydroxyl groups along cellulose chains, we covalently immobilize the fluorescein isothiocyanate (FITC) as indicator and protoporphyrin IX (PpIX) as internal reference onto cellulose acetate (CA), respectively. Subsequently, a series of dual-emission solid fluorescent materials are achieved by simply blending green emitting CA-FITC with red-emitting CA-PpIX with varying ratios. They exhibit a sensitive, color-responsive, rapid and linear response to ammonia in a wide range of 5.0 ppm to $2.5 \times 10^4$ ppm. Benefiting from the excellent solubility and processability of cellulose derivatives, the as-prepared materials are readily processed into different material forms, including printing ink, coating, flexible film, and nanofibrous membrane. The electrospun nanofibrous membrane is successfully employed as a low-cost, high-contrasting, quick-responsive fluorescent trademark for visual monitoring the freshness of shrimp and crab.

---

[1] CAS Key Laboratory of Engineering Plastics and CAS Research/Education Center for Excellence in Molecular Sciences, Chinese Academy of Sciences, 100190 Beijing, China. [2] College of Chemistry, University of Chinese Academy of Sciences, 100049 Beijing, China. [3] CAS Key Laboratory of Organic Solids, Institute of Chemistry, Chinese Academy of Sciences, 100190 Beijing, China. Correspondence and requests for materials should be addressed to J.Z. (email: zhjm@iccas.ac.cn) or to S.W. (email: wangshu@iccas.ac.cn) or to J.Z. (email: jzhang@iccas.ac.cn)

Simple, fast, low-cost and accurate monitoring systems for food safety are becoming more urgent and indispensable, in response to the explosive growth of food safety problems, especially in developing countries. Biogenic amines are considered as important biomarker for monitoring food quality and assistant diagnosis of diseases[1–3], since they usually are generated from the degradation of amino acids due to external microbial activity during food spoilage or endogenous tissue metabolisms[4]. At present, there are many methods for detecting biogenic amines, including liquid chromatography[5], electrochemical devices[6], conductive polymer materials[7], and optical sensors[8]. Among them, the optical sensors have attracted increasing attention, owing to high sensitivity and good selectivity[9]. However, the conventional optical sensors based on the change of the fluorescence intensity of only one luminogen always cannot perform well as expected, because the determination accuracy is easily influenced by the luminogen concentration, instrument and especially external environment, such as temperature, humidity, etc[10]. Moreover, for one luminogen, there is only a minor change of fluorescence intensity in most cases, and the limited sensitivity of human eyes to the change of fluorescence brightness might increase the experimental errors, even cause a failure in the naked-eye detection mode[11]. Furthermore, it is noticeable that the common optical sensors are usually based on the small molecular probes in organic solvents[12], thus they suffer from high toxicity and poor processability. An ideal optical sensor in practice should exhibit not only excellent resolution and accuracy, but also simple operation, short analysis time, and easy visualization[13].

The ratiometric fluorescent systems have shown great potential as ideal optical sensor, for which there always are two luminogens, generally one acts as the indicator of analyte and another as the internal reference[14]. The auto-calibration effect of the incorporated internal reference can significantly improve the detection accuracy based on the strong anti-interference ability[15,16]. More importantly, the sophisticated ratiometric systems established through the rational selection of different luminogens can precisely determine the analyte by the distinct fluorescent color change even with naked-eyes, i.e., visual detection or monitoring[17]. However, the difficulty in synthesis, poor processsibility, high cost, and/or inferior responsiveness still restrict its scope of application in practice.

As the most abundant biopolymer on the earth, cellulose possesses many fascinating properties, such as biocompatibility, biodegradability, inexhaustible renew ability, and environmental friendliness[18]. Taking advantage of the reactive, numerous, and regularly arranged hydroxyl groups along cellulose chains, a wide variety of cellulose derivatives have been developed by simple reactions between functional substituents and active hydroxyls on the cellulose backbone[19–21]. They derive interesting properties from both cellulose and modified functional groups, including good solubility and processability, excellent flexibility, high mechanical strength, and so on. Recently, making full use of the structural characters and reactivity of cellulose skeleton, we successfully convert common aggregation-caused quenching (ACQ) luminogens into excellent solid fluorescent materials through covalently attaching the luminogens onto cellulose skeletons[22]. The synergy between the anchoring and diluting effect on luminogens of cellulose skeleton and the electrostatic repulsion among the ACQ luminogens efficiently inhibits the aggregation and the self-quenching of luminogens.

In this work, we design a cellulose-based ratiometric fluorescent material and propose a visualization method for monitoring the freshness of seafood. Fluorescein isothiocyanate (FITC, green, biogenic-amines indicator) and protoporphyrin IX (PpIX, red, internal reference) are covalently bonded onto cellulose acetate (CA) chains, respectively. Subsequently, the responsive ratiometric fluorescent materials are obtained by blending the cellulose derivatives CA-FITC and CA-PpIX together. The initial fluorescence color is finely tuned according to different blending ratios. The as-prepared ratiometric fluorescent materials not only exhibit fast and reversible response to biogenic amines, but also possess great processability, thus they readily serve as a smart tag to provide practical applications in food industry. Compared with two standard methods for monitoring seafood freshness, including the total volatile basic nitrogen (TVBN) and the colony forming units (CFU), the resultant smart tag provides a high-contrast color change for real-time, visual, and accurate monitoring seafood freshness.

## Results

**Construction of the ratiometric fluorescent materials**. Two common organic fluorophores FITC and PpIX were chosen, where FITC acted as an indicator, while PpIX as an internal reference. In general, due to the planar π-conjugation structures, they suffered from serious aggregation-caused fluorescence quenching (ACQ) phenomenon. Therefore, their emissions were utterly quenched in the solid state[23]. In previous work, our group have developed a universal strategy to fabricate solid fluorescent material with high quantum yield by utilizing the anchoring and diluting effect of cellulose skeleton[22,24,25]. Based on this method, FITC and PpIX were covalently linked to cellulose acetate, respectively. As the images inserted in Fig. 1, green-emitting and red-emitting cellulose-based solid fluorescent materials are obtained.

Notably, the resultant green-emitting CA-FITC and red-emitting CA-PpIX have the partially overlapped excitation bands, indicating that they can be excited at a same wavelength ranged from 350 to 525 nm (Supplementary Fig. 1). Moreover, they exhibit two distinct fluorescence colors according to their completely separated emission bands (475–600 nm and 600–700 nm, respectively). Therefore, by simply mixing these two materials and adjusting their mix ratios, cellulose-based dual-emission fluorescent materials with different initial fluorescence colors, from red to green, are obtained as seen in Fig. 1b.

[1]H-NMR and FTIR spectra of CA-FITC and CA-PpIX (Supplementary Fig. 2 and Supplementary Fig. 3) demonstrate that the two luminogens are successfully bonded on cellulose acetate, owing to the appearance of the characteristic peaks of FITC (6.4–6.8 ppm) and PpIX (6.2–6.7 ppm). However, the contents of luminogens on cellulose acetate are too low to be calculated by [1]H-NMR spectra accurately. Thus, UV-vis spectroscopy was employed to determine the degree of substitution (DS) of luminogens, as shown in Supplementary Fig. 4 and Supplementary Fig. 5. The DS values of FITC and PpIX are 0.0114 and 0.0019, respectively. Considering that the degree of polymerization of cellulose acetate is 112, there is one FITC molecule per polymer chain and one PpIX molecule per every five polymer chains. Compared with physically blending the luminogens in polymer matrix[26], chemical bonding is more efficient to achieve the homogeneous distribution at the molecular scale and relieve their aggregation behavior. Moreover, the chemical immobilizing effect of cellulose acetate could efficiently prevent the migration and leakage of the luminogens during usage, which is crucial to monitor foods.

Due to a negligible amount of luminogens are introduced onto cellulose acetate, the as-prepared materials still exhibit the excellent processability of cellulose derivatives and can be readily transformed into various material forms[27,28]. As shown in Fig. 2, via a solution processing, the cellulose-based dual-emission fluorescent material ($m_{CA-FITC}/m_{CA-PpIX} = 1:5$) with a red initial fluorescence is fabricated into pattern printing, coating on plastic, flexible transparent film, and nanofibrous membrane by

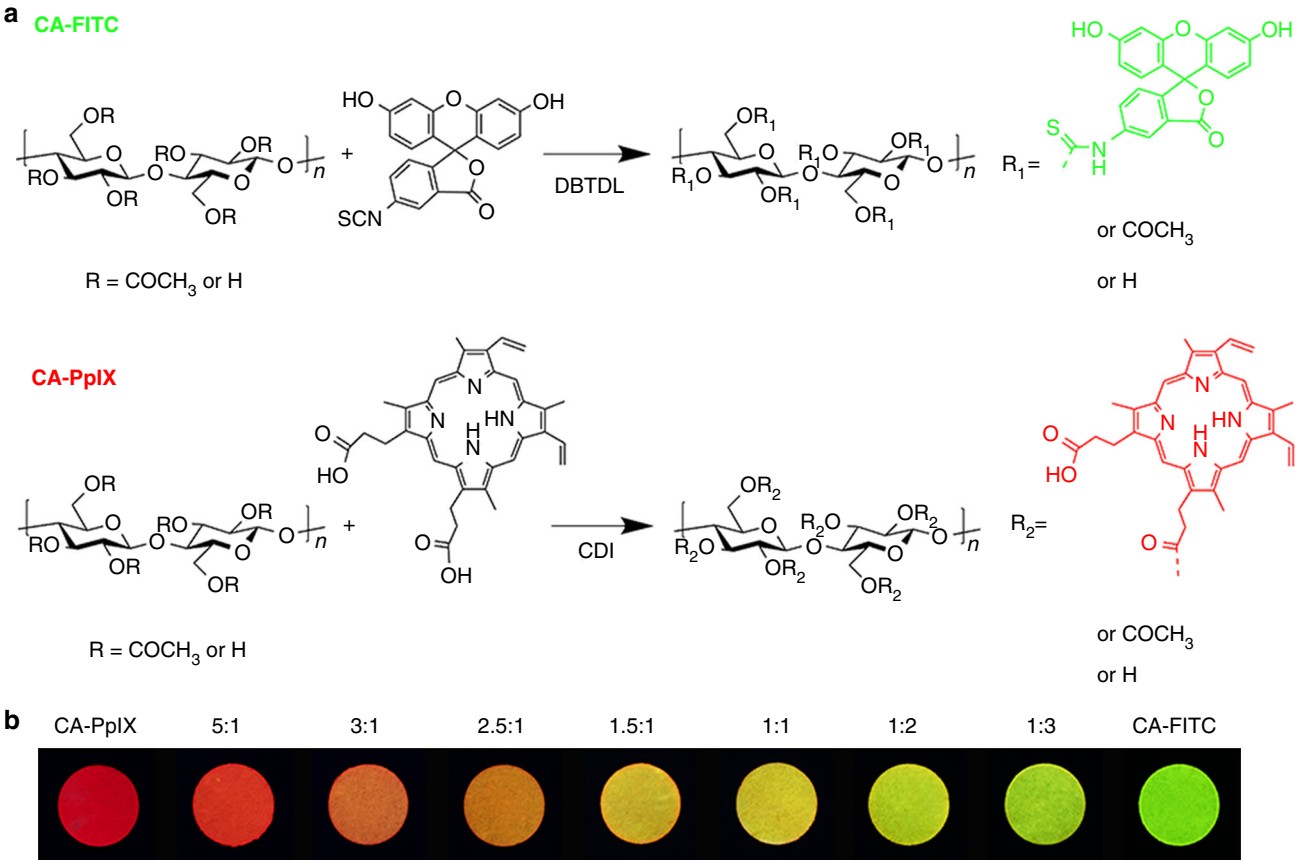

**Fig. 1** Preparation of cellulose-based ratiometric fluorescent materials. **a** Synthetic routes of CA-FITC and CA-PpIX. **b** Cellulose-based dual-emission solid fluorescent materials with different mix ratio of CA-PpIX to CA-FITC (w/w), namely 1:0, 5:1, 3:1, 2.5:1, 1.5:1, 1:1, 1:2, 1:3, 0:1, respectively, under 365 nm UV light

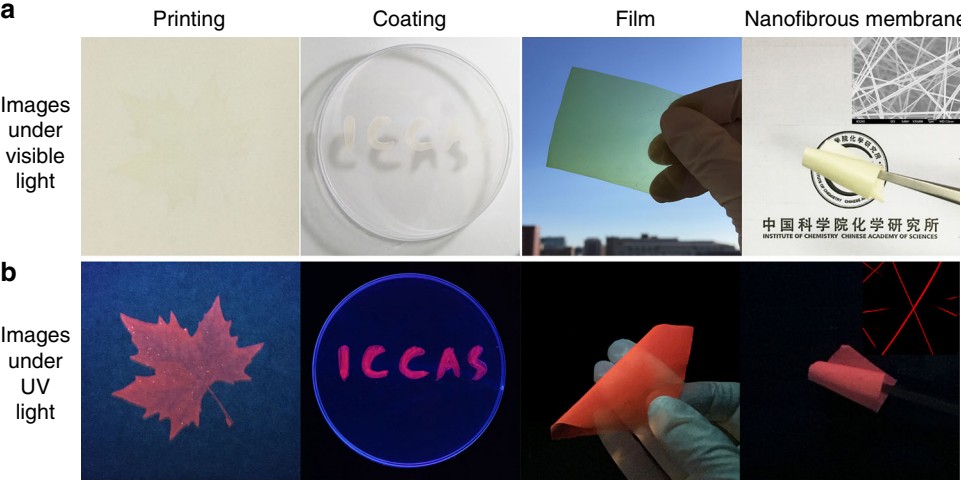

**Fig. 2** Processability of cellulose-based ratiometric fluorescent materials. Images of a cellulose-based dual-emission fluorescent material with a red initial fluorescence in different material forms under **a** visible light and **b** 365 nm UV light. The insets are SEM and CLSM images of nanofibrous membrane, respectively

electrospinning. The images of scanning electron microscopy (SEM) and confocal laser scanning microscopy (CLSM) confirm that the smooth, uniform, and continous nanofibers are formed with a mean diameter of 200 nm. The printing and coating technology make the as-prepared materials very suitable for mass production. Benefiting from the non-toxicity, biodegradability, low cost, and excellent mechanical property of cellulose

acetate[29,30], the obtained dual-emission fluorescent cellulose derivatives are expected to be used as intelligent packaging and smart tag.

**Amine-response behavior**. We used the dual-emission fluorescent cellulose materials as a ratiometric sensor for detecting

ammonia, and the schematic illustration for the sensing mechanism is shown in Fig. 3a. When CA-FITC was blended with CA-PpIX in the ratio of 5:1, the obtained cellulose-based ratiometric fluorescent material exhibited a red fluorescence at the beginning. Upon exposing to ammonia, a proton of FITC was deprived and the molecular structure change resulted in the enhancement of green fluorescence[31], which was confirmed by the change of $^1$H-NMR and UV-vis spectra of FITC (Supplementary Fig. 6). Protoporphyrin IX is irresponsive to ammonia, thus its red fluorescence remained unchanged. With the increase of the ammonia concentration, the green fluorescence of FITC started to be dominant. Thus, an obvious fluorescence color change occurred from red to orange, yellow or green, which was sufficiently clear to be recognized by any untrained observer.

The fluorescence spectra of the above ratiometric fluorescent material with a red initial fluorescence color were shown in Fig. 3b. They exhibits two characteristic emission bands of FITC at 525 nm and PpIX at 670 nm. Upon exposing to ammonia, the fluorescence intensity of FITC increases with the concentration increase of ammonia, while that of PpIX remains constant. In the ammonia concentration range of 5.0 ppm to $2.5 \times 10^4$ ppm, the fluorescence intensity ratio of CA-FITC to CA-PpIX ($I_{FITC}/I_{PpIX}$) is linearly proportional to the logarithmic concentration of ammonia, as seen in Fig. 3c. Based on the calibration equation, we are able to detect ammonia quantitatively. Furthermore, the experiment results demonstrate that along with the increase of the concentration of ammonia, the fluorescence color simultaneously changes from red, to orange, to yellow and to green, which is consistent with our design. More importantly, the response to ammonia of the cellulose-based ratiometric fluorescent material is reversible. After a heating at 60 °C or degassing under vacuum, the fluorescence of FITC quenches, and the fluorescence color returns to its original state, as shown in Fig. 3d. After four cycles, the material still maintains good ammonia responsiveness and high stability. Therefore, the prepared cellulose-based ratiometric fluorescent materials can be re-used for ammonia sensing. This ratiometric approach can effectively eliminate the interference of external environment and processing mode, thus, a more credible and reproducible result can be obtained. Moreover, if a droplet of the ammonia was added on the nanofibers membrane, a rapid response occurred, and the fluorescence color changed instantly (<1 s), as shown in Supplementary Movie 1. Such a sensitive, color change, linearly-dependent and rapid response to ammonia of the cellulose-based ratiometric fluorescent materials indicates that they can be used as a powerful tool to fast and accurately detect ammonia by naked-eyes in real time without the assistance of sophisticated instrument.

The above ratiometric fluorescent material exhibited a fast response to other organic amines also. As shown in Fig. 3e, the fluorescence intensity of the ratiometric fluorescent material gives a distinct change upon contacting with NHEt₂, pyrrolidine, benzylamine, morpholine, putrescine, histamine, hydrazine, and NEt₃, except aniline and urea. In addition, the change degrees of the fluorescence intensity induced by those organic amines are different with each other, because of their different basicity. The $I_{FITC}/I_{PpIX}$ of the control group is about 0.6. At a concentration of amines of 2500 ppm, the $I_{FITC}/I_{PpIX}$ for NH₃ increases to about 5.4, while those for NHEt₂, pyrrolidine, benzylamine, morpholine, putrescine, histamine, hydrazine, and NEt₃ are 4.9, 4.4, 4.3, 4.0, 3.6, 3.5, 3.3, and 1.8, respectively. These organic amines, NH₃, NHEt₂, putrescine, histamine, and NEt₃, are the important components of biogenic amines released by degradation of amino acids, thus, the ratiometric fluorescent materials can be used as an efficient indicator for monitoring metabolic process, such as food spoilage and some diseases occurrence[32,33].

**Monitoring of seafood freshness.** Using shrimps as an example, we tested whether the prepared materials can be used as a smart tag for monitoring the seafood freshness. Red represents fresh, yellow represents slight spoilage, and green represents spoilage. The electrospun membranes consisted of nanofibers are employed for the subsequent test, owing to their large specific surface area and high porosity[34]. As shown in Fig. 4, a piece of electrospun membrane is fixed at the inside top of each packing box of the shrimp which was stored at three different temperatures for 1–5 days. After only one day storage at 25 °C and 4 °C, the fluorescence color of the membrane turned to reseda and yellow, respectively. After five days storage at −16 °C, the fluorescence color of the membrane changed to yellow. As a control, electrospun membranes with wet tissues stored at different temperature for 5 days, and there was no change of the fluorescence color (Supplementary Fig. 7). Except the shrimp, the freshness of crab has been successfully and visually monitored by the smart tag (Supplementary Fig. 8).

In order to confirm the validity of the smart tag, we compared the results obtained from the smart tag with two standard reference methods, the total volatile basic nitrogen (TVBN) and the colony forming units (CFU), which are widely used by food inspection administration[35–37]. We have test the TVBN content in shrimps based on the standard method, as shown in Supplementary Fig. 9. The TVBN scales for raw shrimps are: <12 mg/100 g for fresh, 12–20 mg/100 g for edible but slightly decomposed, 20–25 mg/100 g for borderline, and >25 mg/100 g for inedible and spoiling[35]. The TVBN content in fresh shrimps is about 6.5 mg/100 g. When the shrimps have been stored at 25 °C for 1 day, the TVBN content sharply increases to 132 mg/100 g, which means the shrimps have been totally spoiling. At the same time, the fluorescence color of the smart tag in the package changes from red to bright green, which also indicates the spoilage of the shrimps. When the shrimps have been stored at 4 °C for 1 day, the TVBN content increases to 14 mg/100 g, and the smart tag changes its fluorescence color from red to light yellow. Both the phenomena claim that the shrimps are slightly decomposed. When the shrimps have been stored at 4 °C for 3–5 days, the TVBN content is higher than 24 mg/100 g, and the smart tag changes to green, which clearly demonstrates the spoilage of the shrimps. When the shrimps have been stored at −16 °C for 5 days, the TVBN content slightly increases to about 10 mg/ 100 g, and the fluorescence color of the smart tag changes from red to light yellow, which means the shrimps are slightly decomposed. The TVBN content in crabs was tested too. As shown in Supplementary Fig. 10, the TVBN content in fresh carb is about 5.8 mg/100 g. After storage at 25 °C for 1 day, the TVBN content is about 15 mg/100 g and the smart tag changes to yellow. After storage at 25 °C for 3 days, the TVBN content is about 104 mg/100 g and the smart tag changes to green, which represents the spoilage of the crabs. When the crabs have been stored at 4 °C for 1 day and 3 days, the TVBN contents increase to 8.1 mg/100 g and 21 mg/100 g, respectively. The smart tag changes from red to orange (1 day) and yellow (3 days), which clearly demonstrates the slight decomposition of the crabs. When the crabs have been stored at −16 °C for 3 days, the TVBN content increases gradually to about 14 mg/100 g, and the smart tag changes from red to yellow. These experiment results reveal that the color-change phenomena of the smart tag are completely consistent with the results of the standard TVBN test, so the freshness of the shrimps and crabs can be simply and accurately detected by the smart tag. Taking advantage of this property, the smart tags could be readily applied in the supply chain of shrimps and crabs for real-time and visual monitoring the freshness of shrimps and crabs.

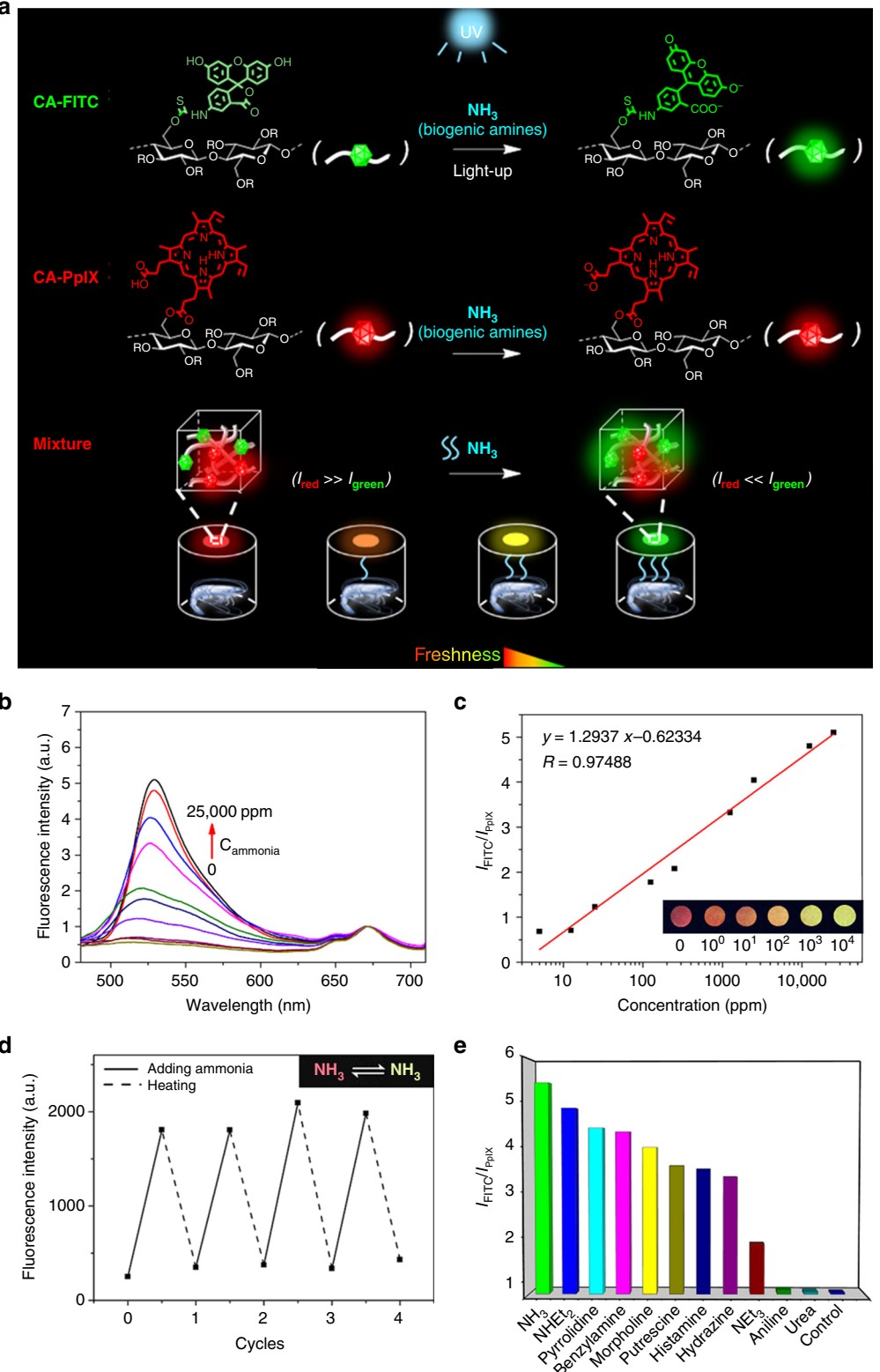

**Fig. 3** Amine-response behavior of cellulose-based ratiometric fluorescent materials. **a** Schematic illustration for the mechanism of the cellulose-based ratiometric fluorescent material. **b** Emission spectra of the nanofibrous membrane dipping in different concentrations of ammonia. **c** Plot of the fluorescence intensity ratio of CA-FITC to CA-PpIX ($I_{FITC}/I_{PpIX}$) versus concentration of ammonia. Inset: fluorescence images of the membrane treated with a series of different concentrations of ammonia. **d** Fluorescence enhancement and quenching cycles of the emission peak at $\lambda_{em} = 525$ nm upon the nanofibrous membrane dipping in 2500 ppm ammonia and following heated. Inset: reversible color change of the nanofibrous membrane in a cycle. **e** Fluorescence response of the nanofibrous membrane to ammonia and other organic amines in water (2500 ppm). Excitation wavelength, 365 nm

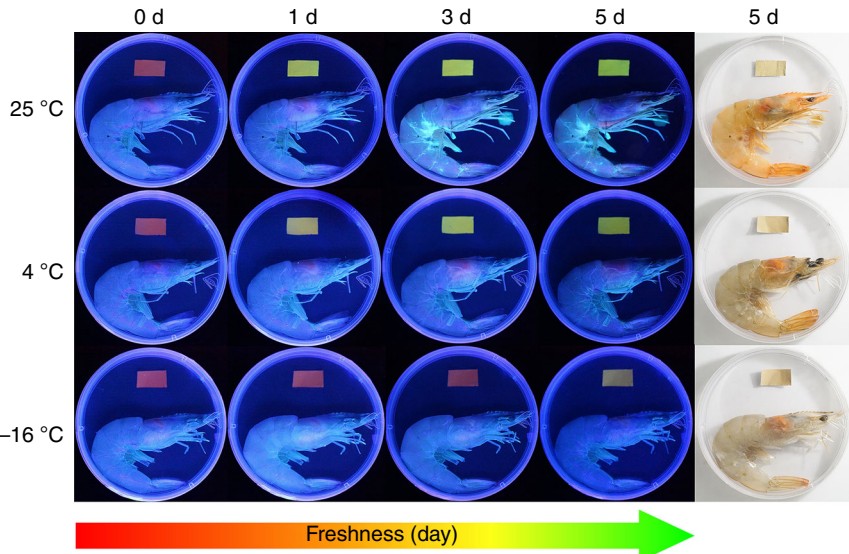

**Fig. 4** Monitoring of shrimp freshness by the smart trademark. Images of the cellulose-based ratiometric fluorescent material with a red initial fluorescence as a smart trademark for monitoring the freshness of shrimps stored at different conditions

The colony forming units (CFU) in shrimps and crabs were also measured for comparison. Actually, it's hard to determine the CFU limit value which represents the total spoilage of shrimps or crabs, because original CFU of shrimps and crabs is subjected to various factors, such as species, producing area and fishing time. Therefore, there is no national standard about this issue. But, the change of CFU value during storage can be generally used to evaluate the freshness of shrimps and crabs. A significant increase of CFU means the decrease of freshness[38,39]. The trend of the CFU results is in accord with that of the TVBN results, as shown in Supplementary Fig. 11 and Supplementary Fig. 12. When the fresh shrimps have been stored at 25 °C for 1 day, the CFU value grows exponentially, so the freshness decreases sharply[40,41]. When the shrimps have been stored at 4 °C, the CFU value decreases at first because of the death of the chill-intolerant microorganism, then significantly increases, which means the freshness begin to decrease. When the shrimps have been stored at −16 °C for 1–7 days, the CFU value also dramatically decreases at first, then increases slowly. The increase of bacteria means a decrease of the freshness. Comparing with the CFU value in the shrimps stored at 4 °C, the CFU value at −16 °C remains at a lower level. Thus, the shrimps are just slight decomposed after 5 days. The change rule of CFU in crabs is similar with that of shrimps. When the crabs have been stored at 25 °C, the CFU value increases sharply, as a result, the freshness of crabs decreases. When the crabs have been stored at 4 °C, the CFU decreases at first and increases after 3 days, so, after a 3-day storage, the freshness of crabs decreases. Therefore, the smart tags could be readily applied in the supply chain of shrimps and crabs for real-time and visual monitoring the freshness of shrimps and crabs.

## Discussion

Overall, simple, fast, and accurate detection of food freshness has great significance to ensure food safety, which provides practical applications in food business. We developed a cellulose-based ratiometric fluorescent material with superior amine-response by covalently linking FITC and PpIX luminogens onto cellulose. Green-emitting CA-FITC served as a biogenic-amines indicator, while red-emitting CA-PpIX as an internal reference. The as-prepared material exhibited a sensitive, color change, linearly dependent, reversible, and rapid response to biogenic amines,

including $NH_3$, pyrrolidine, benzylamine, morpholine, putrescine, histamine, hydrazine, and $NEt_3$. Upon contacting with biogenic amines, particularly ammonia, their fluorescence color immediately changed from red to orange or green, and the fluorescence intensity ratio of CA-FITC to CA-PpIX ($I_{FITC}/I_{PpIX}$) was linearly proportional to the logarithmic concentration of ammonia in a wide range of 5.0 ppm to $2.5 \times 10^4$ ppm. Such a promising responsiveness offers a great opportunity to fast and accurately detect biogenic amines by naked-eyes in real time. Besides, taking advantages of the good solubility, processability, and biocompatibility of cellulose derivatives, the ratiometric fluorescent material can be fabricated into many material forms, such as intelligent package and smart tag. The electrospun membrane was employed as a simple, low-cost, high-contrasting, quick-responsive, and smart trademark for visual monitoring the freshness of shrimp and crab successfully, which was validated against the standard food-monitoring methods, TVBN, and microbial activity. These materials have great significance to ensure food safety and provide practical applications in food business.

## Methods

**Materials**. Cellulose acetate (Mn = 30000, DS = 2.5) was used after drying in the vacuum oven at 70 °C for 24 h caisothiocyanate (FITC), protoporphyrin IX (PpIX), dibutyltindilaurate (DBTDL), N,N'-carbonyldiimidazole (CDI) were purchased from J& K Scientific Ltd. and used as received. N,N'-Dimethylformamide (DMF) was dehydrated by $CaH_2$ and preserved with 3 Å molecular sieve after distillation. Super-dry dimethyl sulfoxide (DMSO) with molecular sieves was purchased from J& K Scientific Ltd. and used as received. Ammonia ($NH_3$, 25% in water), diethylamine ($NHEt_2$), triethylamine ($NEt_3$), and urea were purchased from Beijing Chemical Works and used as received. Aniline, benzylamine, hydrazine (64%), and histamine were purchased from J& K Scientific Ltd. and used as received. Morpholine, pyrrolidine, and putrescine were purchased from Sigma-Aldrich, Inc. and used as received.

**Synthesis of CA-FITC**. The synthesis was performed in the Schlenk line. Firstly, 3 mmol cellulose acetate was dissolved in 20 mL of DMF at 80 °C to get a homogeneous solution. Then, 0.1 mmol FITC and 1.5 mmol DBTDL as catalyst were added into the cellulose acetate solution followed by stirring at 100 °C for 4 h. The product was precipitated in the mixed solvent of methanol and water (v/v = 1/1). The generated precipitation was filtered and further washed three times followed by drying in the vacuum oven at 80 °C for 24 h to offer a yellow powder.

**Synthesis of CA-PpIX**. Cellulose acetate (4 mmol) was dissolved in 20 mL of DMSO. In another flask, 0.0667 mmol PpIX was activated by CDI. Then, the prepared PpIX solution was added into the solution of cellulose acetate. The

obtained mixture was stirred at 80 °C for 20 h. The final product was precipitated in ethanol and further washed three times followed by drying in the vacuum oven at 80 °C for 24 h to offer a brown powder.

**Preparation of ratiometric fluorescent materials**. A homogeneous solution was prepared by dissolving CA-FITC and CA-PpIX in DMF with a mass ratio of 1:3. By using EPSON ME-10 printer and replacing the printing ink with the solution of CA-FITC and CA-PpIX, a designed pattern was obtained. Coating on plastic was performed by writing with Chinese brush. The transparent film was prepared by casting the solution on the glass and volatilizing the solvent. The nanofibrous membrane was fabricated by electrospinning. Firstly, a solution was prepared by dissolving CA-FITC and CA-PpIX in DMAc/acetone (v/v = 1/2) at the concentration of 15 wt%. Then, the solution in the syringe was electrospun at a voltage of 20 kV with a tip-to-collector distance of 15 cm and a flow rate of 0.6 mL/h to afford a nanofibrous membrane. DW-P303-0.1 AC high voltage power supply was used as the power source, and LSP01-1A from LongerPump was used as the injection pump.

**Preparation and detection of amine solutions**. Commercial ammonia was diluted with different quantity of deionized water to obtain a series of ammonia solutions with known concentration. Then, the nanofibrous membrane was immersed in the ammonia solution for 3 min. After being taken out from the solution, the fluorescence of the nanofibrous membrane was recorded as soon as possible. The preparation and detection process of other amines solutions were the same as that of ammonia.

**Detection of the total volatile basic nitrogen**. According to the Chinese Standard GB 5009.228-2016, the total volatile basic nitrogen (TVBN) in shrimps and crabs was determined. All the shrimps were removed the shell and head. The crabs were removed the shell. The meat of shrimp and carb was used for the experiments. All experiments were performed in triplicate.

**Detection of microbial count in shrimps and crabs**. According to the Chinese Standard GB 4789.2-2010 for food safety, the colony forming units (CFU) in shrimps and crabs are determined. Each sample was derived from three shrimps or three crabs in the same storage environment and was given an alcohol sterilization before detection. The final experiment results were obtained by plate counting method.

**Characterization**. $^1$H-NMR spectrum was measured on Bruker Avance 400 M NMR spectrometer. Lambda 35 spectrophotometer from Perkin–Elmer was used to record UV-vis absorption. Fluorescence emission and excitation spectra were measured on HITACHI F-4500 fluorescence spectrometer. The morphology of the nanofibrous membrane was investigated by SEM (JSM-6700F) and CLSM (OLYMPUS, FV1000-IX81). Thermogravimetric analyzer TGA 8000 from Perkin–Elmer was used to record the TGA curves of CA-FITC and CA-PpIX under nitrogen atmosphere. The FTIR spectra were measured by NICOLET 6700 from Thermo-Fisher Scientific. Differential scanning calorimetry (DSC) was conducted on TA-Q200 differential scanning calorimeter (TA Instruments, USA) under nitrogen atmosphere. The total volatile basic nitrogen (TVBN) was measured by K-375 Kjeldahl apparatus from BUCHI Corporation.

## Data availability

All relevant data are included in this Article and its Supplementary Information files. The freshness detection data have been submitted to https://figshare.com/. https://doi.org/10.6084/m9.figshare.7545638.

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

## Acknowledgements

This work was supported by the National Science Foundation of China (Nos. 51425307, 51573196, 21533012, and 91527306), the Program of Taishan Industry Leading Talents (Shandong Province), and the Youth Innovation Promotion Association CAS (No. 2018040).

## Author contribution

R.J., J.Z., S.W., and J.Z. conceived the idea. R.J. performed the experiments. W.T. and H. B. offered help to R.J. for the experiments. R.J., W.T., J.Z., S.W., and J.Z. discussed the results and wrote the manuscript.

## Additional information

**Competing interests:** The authors declare no competing interests.

