## [Peer Review File · Nature Communications]

Reviewers' Comments:

Reviewer #1:

Remarks to the Author:

This is a timely contribution to the sensor world. The work has been carried out with care and both the manuscript and SI are well written. The problem is that this reviewer is just not convinced that either the approach or the results warrant publication in Nature Comm. The idea of modulating cellulose is well known, as the idea of ratiometric probe development. Thus, in essence, the present submission simply combines known approaches to detect a series of substrates (amines) that have often been the target of the sensor development community.

I know this is sad news to the authors, but targeting a more appropriate (and less high profile) journal should see this work accepted without difficulty. It would certainly see nice reviews were if to come back to me as a referee at a less-high profile journal.

The above lack of full enthusiasm notwithstanding, I would certainly not object if the other referees are much more favorable and the editors elect to proceed toward publication at Nature Comm.

Reviewer #2:

Remarks to the Author:

The manuscript by Jia et al. describes a novel cellulose-based ratiometric fluorescent material with superior amine-response by covalently linking FITC and PpIX luminogens onto cellulose, which shows a sensitive, color-change, linearly-dependent, reversible and rapid response to biogenic amines. Furthermore, the electrospun membrane has been successfully employed as a quick-responsive and smart trademark for visual monitoring the freshness of shrimp. This manuscript is interesting and well-prepared, and I recommend publication of this manuscript in Nature Communications after minor revisions. The comments are as follows.

Comments to the Authors:

1. Please provide melting point, IR and TGA data for CA-FITC and CA-PpIX.
2. How about the influence of benzylamine, hydrazine, putrescine, histamine, and bulky cyclic amines (such as pyrrolidine, piperidine and morpholine) ?
3. Please provide the ¹H NMR and UV-vis spectra changes of CA-FITC in DMSO upon the addition of ammonia to support the sensing mechanism shown in Figure 3a.
4. Line 111 and 117, the order of Figs. S1 and S2 are wrong.

Reviewer #3:

Remarks to the Author:

The authors present a research paper about the design and preparation of the cellulose-based ratiometric fluorescent materials responding to ammonia and some other (biogenic) amines. The approach of preparing the fluorescent materials for food safety detection is interesting and could really be promising when preparing highly stable sensing materials with no dye leaching, which is one of the crucial requirements when sensing food. However, I have some concerns based on which this manuscript cannot be published in the way as it is presented now and should require additional experimental data to deserve a publication in a journal like Nature Communications. The manuscript could be improved with respect to the following issues:

Major comments:

- Additional experiments should be done to determine the exact freshness of the shrimps. The statement in line 194 of the manuscript ('Red represents fresh, while yellow or green represents spoilage') is not supported with any data of microbial population evaluation of shrimps based on which the authors could estimate/determine with a more reliable probability what is the freshness of the shrimps. The authors should compare the microbial population (which increase during seafood storage – depending on the storage temperature and time) with the signal of their fluorescent sensor material in order to validate the performance of their fluorescent material. Namely, many seafood as well as fish, being still fresh, have their initial amine-like odour and the ammonia/amine sensitive sensor will respond to these gasses as soon as exposed to these foods – this does not mean that the food tested is already spoiled after, for instance, one day of storage at 5°C. I recommend the authors read the article in *Sensing and Bio-Sensing Research* 6 (2015) 28-32 ...
- The authors should test a variety of seafoods, not only shrimps. This would give a broader overview of the performance of their fluorescent materials, which they call smart trademark that can be widely used in the supply chain of seafood
- Figure 3a-d): It is not clear how authors have determined/controlled the concentration of ammonia, when evaluating the sensing response of their cellulose-based fluorescent materials. Did you have a reference electrode/sensor? Should be explained in the experimental part ...
- Figure 3a-d): In which form the ammonia gas was added near/onto (?) the sensing material? As a drop of concentrated ammonium hydroxide solution or injected as a gas from the ammonia cylinder? Again, how did you controlled the ammonia concentration? The exact procedure should be explained in the experimental part.
- Figure 3 e): what is the concentration of other amines that the fluor. materials were exposed to? How did you defined the concentration (2500 ppm valid for all analytes?) of other amines added to the fluorescent materials?
- It is not clear how different fluorescent materials were prepared into pattern printing, coating on plastic, film etc.. There is written nowhere in the experimental part, which instrumentation, machinery or equipment the authors used to prepare those materials. Except in the Fig. 4 and Fig. S5, there is also not clear in which experiment is used which material.
- What is the response time of the fluorescent materials upon exposure to ammonia/amines... several seconds, minutes ? Exact time should be given for each tested analyte ...

Minor comments:

- Line 111 and 117: there is a mistake in citing the Figures S1 and S2, should be the opposite ...
- Figure 2 should be labelled as a) (images under visible light) and b) (images under UV light); a) and b) should be added in the Figure caption too.
- Line 151: in which form is the mentioned material that is supposed to be also on Figure 3a)? ... film, powder, nanofibrous membrane ...?
- Line 182: please add corresponding reference(s) at the end of the sentence '...such as food spoilage and some disease occurrence.'
- Line 186: brackets for the reference 32 are missing.
- The manuscript should undergo English proofreading.

Other comments in the Supporting information:

- Line 17: there are not written in the experimental part the exact names of amine chemicals (along with %, if it was a solution) and from which company they were provided. Should be added ...
- Equipment for the preparation of pattern printing, coating and transparent film, electrospinning, ... should be added ...
- Line 43: which type of UV-Vis spectrophotometer from Perkin-Elmer was used?
- Fig. S3 and S4: there are missing the concentrations units ...
- Fig. S5: add the exact number of days of exposure in the fig. caption ...

Responses to the reviewers' comments and list of the details of the changes:

1. Answers to Comments by Reviewer # 1:

- (1) **Reviewer # 1 wrote:** *This is a timely contribution to the sensor world. The work has been carried out with care and both the manuscript and SI are well written. The problem is that this reviewer is just not convinced that either the approach or the results warrant publication in Nature Comm. The idea of modulating cellulose is well known, as the idea of ratiometric probe development. Thus, in essence, the present submission simply combines known approaches to detect a series of substrates (amines) that have often been the target of the sensor development community. I know this is sad news to the authors, but targeting a more appropriate (and less high profile) journal should see this work accepted without difficulty. It would certainly see nice reviews were it to come back to me as a referee at a less-high profile journal. The above lack of full enthusiasm notwithstanding, I would certainly not object if the other referees are much more favorable and the editors elect to proceed toward publication at Nature Comm.*

Answer: Thanks for your kind suggestion. The novelty of this work includes three features: (1) In this paper, we designed and prepared natural cellulose-based ratiometric fluorescent materials with superior amine-response, which offered real-time and visual detection of food safety. (2) Furthermore, benefiting from the excellent solubility and processibility of cellulose derivatives, the as-prepared materials were readily processed into different formats, including printing ink, coating, flexible film, and nanofibrous membrane. (3) Our experiment showed that electrospun nanofibrous membrane could be used as a simple, low-cost, high-contrasting, quick-responsive fluorescent trademark for visual monitoring the freshness of shrimps and other seafood. **The other two referees strongly support this work to publish at this journal.**

2. Answers to Comments by Reviewer # 2

- (1) **Reviewer # 2 wrote:** *Please provide melting point, IR and TGA data for CA-FITC and CA-PpIX.*

Answer: Thanks for your kind suggestion. The IR, DSC and TGA data of CA-FITC and CA-PpIX were provided in the revised supporting information, as shown in Fig. S3. CA-FITC and CA-PpIX will degrade before melting, hence, there is no melting point. Even so, differential scanning calorimeter (DSC) was used to evaluate the thermal property of CA-FITC and CA-PpIX, and their glass transition temperatures (T_g s) were shown in Fig. S3.

- (2) **Reviewer # 2 wrote:** *How about the influence of benzylamine, hydrazine, putrescine, histamine, and bulky cyclic amines (such as pyrrolidine, piperidine and morpholine)?*

Answer: We have evaluated the influence of benzylamine, hydrazine, putrescine, histamine, and bulky cyclic amines on the fluorescence change of the ratiometric

fluorescent material. The results have been added into Fig. 3e. A detailed description has been made in the revised manuscript as well.

(3) Reviewer # 2 wrote: *Please provide the ^1H NMR and UV-vis spectra changes of CA-FITC in DMSO upon the addition of ammonia to support the sensing mechanism shown in Figure 3a.*

Answer: Thanks for your kind suggestion. Because the content of FITC on cellulose acetate was very low (DS = 0.0114), the ^1H NMR spectra change of CA-FITC upon the addition of ammonia is obscure. In order to support the sensing mechanism, we tested the ^1H NMR and UV-vis spectra of FITC upon the addition of ammonia. The results were shown in Fig. S6. It can be seen that, after the addition of ammonia, the peak of FITC at 11 ppm which was attributed to the active hydrogen disappears, and most of other peaks of FITC shift high-field markedly. The UV-vis spectra exhibited significant change upon addition of ammonia also. Both ^1H NMR and UV-vis spectra indicate the molecular structure change of FITC. A proton of FITC was deprived, as shown in Fig. 3a and S6.

(4) Reviewer # 2 wrote: *Line 111 and 117, the order of Figs. S1 and S2 are wrong.*

Answer: Thanks for your kind suggestion. We have revised the error in the revised manuscript.

3. Answers to Comments by Reviewer # 3

(1) Reviewer 3 # wrote: *Additional experiments should be done to determine the exact freshness of the shrimps. The statement in line 194 of the manuscript ('Red represents fresh, while yellow or green represents spoilage') is not supported with any data of microbial population evaluation of shrimps based on which the authors could estimate/determine with a more reliable probability what is the freshness of the shrimps. The authors should compare the microbial population (which increase during seafood storage – depending on the storage temperature and time) with the signal of their fluorescent sensor material in order to validate the performance of their fluorescent material. Namely, many seafood as well as fish, being still fresh, have their initial amine-like odour and the ammonia/amine sensitive sensor will respond to these gasses as soon as exposed to these foods – this does not mean that the food tested is already spoiled after, for instance, one day of storage at 5 ° C. I recommend the authors read the article in Sensing and Bio-Sensing Research 6 (2015) 28-32.*

Answer: Thanks for your kind suggestion. We have seriously read the article of Sensing and Bio-Sensing Research 6 (2015) 28-32. Microbial population evaluation of the shrimps under different storage conditions has been tested by using the standard method (Chinese Standard GB 4789.2-2010), and the results were shown in Fig. S9. It is clear that the colony forming units (CFU) in shrimps stored at 25 °C increases sharply with the increase of the storage days. The CFU in shrimps stored at 4 °C reduces at first, and then increases to twice as much as the original CFU in the fresh shrimps. The initial decrease of the microbial population is attributed to the death of

the cold intolerant bacteria. Similarly, the CFU in shrimps stored at -16 °C reduces at first, then increases, and the highest CFU is less than 30% of the original CFU in the fresh shrimps. The above results reveal that the shrimps stored at -16 °C could more effectively inhibit the microbial population. Except bacteria, enzymes can also lead to the spoilage of the food during transportation and storage. The shrimps and dead bacteria will be decomposed by enzymes at suitable conditions, such as temperature and humidity. During this process, the amines are produced and released. Hence, it's more accurate and reliable to detect the freshness of seafoods by measuring amines rather than only monitoring microbial population, which is time-consuming and destructive. However, the accessory microbial population experiment fosters a deeper understanding of the spoilage process for the readers.

In our work, we actually focused on fabricating a rapid, safe, non-destructive, visual, biodegradable and cheap fluorescent tag to distinguish the freshness of seafoods by ordinary people. We can also monitor the throughout seafood supply chain, "from farm to table". Furthermore, the initial amine-like odor will not change the fluorescence color of the smart tag, because the fresh shrimps also have the amine-like odor. When the concentration of the amines is above the critical value, the fluorescence color change of the tag occurs, which means the shrimps is not fresh.

(2) Reviewer 3 # wrote: *A variety of seafoods, not only shrimps, should be tested to give a broader overview of the performance of the fluorescent materials.*

Answer: Thanks for your kind suggestion. Crab and seabass have been tested to verify the performance of the fluorescent material as smart tag. The experiment results have been given in Fig. S10 and S11.

(3) Reviewer 3 # wrote: *It is not clear how to determine/control the concentration of ammonia, when evaluating the sensing response of their cellulose-based fluorescent materials.*

Answer: The smart tag is expected to be placed in the package of seafood which is filled with water or water vapor, so we measured the fluorescence response to ammonia in aqueous solution. The commercial ammonia was diluted with deionized water to obtain a series of ammonia solutions with accurate concentrations. These information has been added in the experimental part.

(4) Reviewer 3 # wrote: *In which form the ammonia gas was added near/onto (?) the sensing material?*

Answer: The sensing membrane was immersed in the ammonia solution for 3 mins, or the ammonia solution was added onto the sensing material. Because taking a fluorescence spectrum needs at least 20 s, the first method (immersion method) was employed. After being taken out from the solution, the fluorescence spectrum of the sensing material was recorded as soon as possible. In addition, the ammonia/water gas was used to confirm the responsivity of the sensing materials. The ammonia/water gas was obtained by heating the ammonia solution, then was piped to the surface of the

sensing materials. In this case, it is difficult to control the concentration of NH_3 . So, we obtained the detection limit by using the solution method.

(5) Reviewer 3 # wrote: *What is the concentration of other amines that the fluorescence materials were exposed to? How did you define the concentration (2500 ppm valid for all analytes?) of other amines added to the fluorescent materials?*

Answer: It was the same as ammonia. The commercial amines were diluted with deionized water to obtain a solution with a concentration of 2500 ppm.

(6) Reviewer 3 # wrote: *It is not clear how different fluorescent materials were prepared into pattern printing, coating on plastic, film etc.*

Answer: Thanks for your kind suggestion. We have added the detailed preparation process in the experimental part.

(7) Reviewer 3 # wrote: *What is the response time of the fluorescent materials upon exposure to ammonia/amines ... several seconds, minutes? Exact time should be given for each tested analyte ...*

Answer: We have added a video in the supporting information (Fig. S7). The response time is very short (< 1 s). After a droplet of the ammonia being added onto the surface of the nanofibers membrane, the fluorescence color changed instantly.

(8) Reviewer 3 # wrote: *Line 111 and 117: there is a mistake in citing the Figures S1 and S2.*

Answer: Thanks for your kind suggestion. We have revised this error in the revised manuscript.

(9) Reviewer 3 # wrote: *Figure 2 should be labelled as a) (images under visible light) and b) (images under UV light); a) and b) should be added in the Figure caption too.*

Answer: Thanks for your kind suggestion. We have added this information in the revised manuscript.

(10) Reviewer 3 # wrote: *In which form is the mentioned material that is supposed to be also on Figure 3a)? ... film, powder, nanofibrous membrane ... ?*

Answer: All of the material forms of the sensing materials, including film, powder, coating and nanofibrous membrane, exhibit a good response to ammonia. The material form with the highest specific surface area gives the best detection result. Thus, the nanofibrous membrane is the best choice.

(11) Reviewer 3 # wrote: *Please add corresponding reference(s) at the end of the sentence '...such as food spoilage and some disease occurrence.*

Answer: Thanks for your kind suggestion. We have added references as your

suggestion.

(12) Reviewer 3 # wrote: *Brackets for the reference 32 are missing.*

Answer: Thanks for your kind suggestion. The notation of references has been revised to the format demanded by Nature Communication.

(13) Reviewer 3 # wrote: *The manuscript should undergo English proofreading.*

Answer: Thanks for your kind suggestion. As your suggestion, our manuscript have been revised seriously by ACS Language Editors.

(14) Reviewer 3 # wrote: *There are not written in the experimental part the exact names of amine chemicals (along with %, if it was a solution) and from which company they were provided.*

Answer: Thanks for your kind suggestion. Detailed information has been added in the experimental part.

(15) Reviewer 3 # wrote: *Equipment for the preparation of pattern printing, coating and transparent film, electrospinning, should be added.*

Answer: Thanks for your kind suggestion. Detailed information has been added in the experimental part.

(16) Reviewer 3 # wrote: *Line 43: which type of UV-Vis spectrophotometer from Perkin-Elmer was used?*

Answer: Lambda 35 spectrophotometer from Perkin-Elmer was used to record UV-vis absorption. Detailed information has been added in the experimental part.

(17) Reviewer 3 # wrote: *Fig. S3 and S4: there are missing the concentrations units.*

Answer: Thanks for your kind suggestion. We have added the concentration unit in the revised manuscript.

(18) Reviewer 3 # wrote: *Fig. S5: add the exact number of days of exposure in the figure caption.*

Answer: Thanks for your kind suggestion. We have revised the figure caption in the revised manuscript.

Reviewers' Comments:

Reviewer #1:

Remarks to the Author:

This is substantially improved submission. This referee was lukewarm at the time of the initial review. However, if the authors' statement about the other referees being enthusiastic is in fact true, the promise of supporting publication will be kept. There is still too much hype and the conclusion of this being widely applicable as a probe for seafood freshness is not supported by enough examples to be considered credible. The sea bass experiments give rise to a barely discernible change (compare Fig. S11 with Fig. S11) and that was the only non-crustacean tested. I think with a change in title to "shrimp and crab" instead of "seafood", this paper can be accepted as is.

Reviewer #2:

Remarks to the Author:

Now, the manuscript has been well-revised. I recommend publication of this manuscript in Nature Communications.

Reviewer #3:

Remarks to the Author:

Dear Authors,

After reading the responses to all reviewers' comments my final decision is I do not recommend the publication of the manuscript in a such high profile journal like Nature Communications. Please find my comments bellow:

- I have never written, that I 'strongly support your work to be published in Nature Communications', but, rather, 'additional experimental data is required to deserve a publication in a journal like Nature Communications.'

- The statement that your method 'is more accurate and reliable to detect the freshness of seafoods by measuring amines rather than only monitoring microbial population, which is time-consuming and destructive' I could agree with, but whenever a new type of sensing probe/sensor is being developed, researchers should validate their new probe/method against some reference material/method; this could be done by measuring relevant bacteria (in this case) and, of course, by comparing the results obtained with the new probe with some standard reference method – only then the authors could talk about the accuracy and reliability of their new method. This step (comparing the concentration of the analyte measured with the new probe and the reference method) is usually never skipped when evaluating the performance of the new probe, and this is actually one of the analytical aspects that should be considered in sensor development.

- I am not satisfied with the set of other tested sea food, as there should be much more included. To my experience the shrimps is the worse choice to test because they are very easily spoiled and they have very strong initial odour. On the other hand, tuna is also a type of fish that is very sensitive in freshness; tuna could be not fresh any longer, but there is no amine-like odour released from it. Based on only three different seafoods and without validation and comparison with a reference method, authors cannot say about the real freshness of the seafoods and reliability of the results obtained. Authors also should define which seafood their method is appropriate for in order to 'readily provide practical applications of their sensors in food industry'.

- Finally, I also agree with the first reviewer about the well known ideas on cellulose modulating and ratiometric measurement principle.

I am sorry to say that based on the above raised issues, this work in the present state should be more appropriate for publication in another relevant journal with lower impact factor.

Responses to the reviewers' comments and list of the details of the changes:

1. Response to Comments by Reviewer # 1:

Reviewer # 1 wrote: *This is substantially improved submission. This referee was lukewarm at the time of the initial review. However, if the authors' statement about the other referees being enthusiastic is in fact true, the promise of supporting publication will be kept. There is still too much hype and the conclusion of this being widely applicable as a probe for seafood freshness is not supported by enough examples to be considered credible. The sea bass experiments give rise to a barely discernible change (compare Fig. S11 with Fig. S11) and that was the only non-crustacean tested. I think with a change in title to "shrimp and crab" instead of "seafood", this paper can be accepted as is.*

Answer: Thanks for your kind suggestion. We have revised the title as your kind suggestion.

2. Response to Comments by Reviewer # 2

Reviewer # 2 wrote: *Now, the manuscript has been well-revised. I recommend publication of this manuscript in Nature Communications.*

Answer: We sincerely appreciate your recommendation.

3. Response to Comments by Reviewer # 3

(1) Reviewer 3 # wrote: *The statement that your method 'is more accurate and reliable to detect the freshness of seafoods by measuring amines rather than only monitoring microbial population, which is time-consuming and destructive' I could agree with, but whenever a new type of sensing probe/sensor is being developed, researchers should validate their new probe/method against some reference material/method; this could be done by measuring relevant bacteria (in this case) and, of course, by comparing the results obtained with the new probe with some standard reference method – only then the authors could talk about the accuracy and reliability of their new method. This step (comparing the concentration of the analyte measured with the new probe and the reference method) is usually never skipped when evaluating the performance of the new probe, and this is actually one of the analytical aspects that should be considered in sensor development.*

Answer: Thanks for your suggestion. In order to confirm the validity of our smart tag, we compared the results obtained from the smart tag with two standard methods, the total volatile basic nitrogen (TVBN) and the colony forming units (CFU), which are widely used by food inspection administration (The Hygienic Standard for Fresh and Frozen Aquatic Products, *National Standard GB 2733-2015 of the People's Republic of China*; Huss H.H., Quality and quality changes in fresh fish (*Food and Agriculture Organization of the United Nations*); Okpala COR, Choo WS, Dykes GA, Quality and shelf life assessment of Pacific white shrimp (*Litopenaeus vannamei*) freshly harvested and stored on ice, *LWT - Food Science and Technology*, 2014, 55, 110-116; Bazemore R,

中国科学院化学研究所

INSTITUTE OF CHEMISTRY, THE CHINESE ACADEMY OF SCIENCES

Fu SG, Yoon Y, Marshall D, Major Causes of Shrimp Spoilage and Methods for Assessment, *American Chemical Society: Washington, DC, 2003*).

We have test the TVBN content in shrimps based on the standard method, as shown in **Figure 1**. The TVBN scales for raw shrimps are: <12 mg/100 g for fresh, 12-20 mg/100 g for edible but slightly decomposed, 20-25 mg/100 g for borderline, and >25 mg/100 g for inedible and spoiling (Okpala COR, Choo WS, Dykes GA, LWT - Food Science and Technology, 2014, 55, 110-116). The TVBN content in fresh shrimps is about 6.5 mg/100 g. When the shrimps have been stored at 25 °C for 1 day, the TVBN content sharply increases to 132 mg/100 g, which means the shrimps have been totally spoiling. At the same time, the fluorescence color of the smart tag in the package changes from red to bright green, which also indicates the spoilage of the shrimps. When the shrimps have been stored at 4 °C for 1 day, the TVBN content increases to 14 mg/100 g, and the smart tag changes its fluorescence color from red to light yellow. Both the phenomena claim that the shrimps are slightly decomposed. When the shrimps have been stored at 4 °C for 3-5 days, the TVBN content is higher than 24 mg/100 g, and the smart tag changes to green, which clearly demonstrates the spoilage of the shrimps. When the shrimps have been stored at -16 °C for 5 days, the TVBN content slightly increases to about 10 mg/100 g, and the fluorescence color of the smart tag changes from red to light yellow, which means the shrimps are slightly decomposed. These experiment results reveal that the color-change phenomena of the smart tag are completely consistent with the results of the standard TVBN test, so the shrimp freshness can be accurately detected by our smart tag. Moreover, our smart tag has provided a simple and more sensitive approach to assess the fresh state of shrimps before totally spoilage, i.e. slightly decomposed. Taking advantage of this property, our smart tags could be readily applied in the shrimp supply chain for real-time and visual monitoring the freshness of shrimps.

Figure 1. (a) The TVBN values in shrimps stored at 25 °C, 4 °C and -16 °C for different days; (b-d) The TVBN values in shrimps stored at 25 °C, 4 °C and -16 °C, respectively.

The colony forming units (CFU) in shrimps were also measured for comparison. The trend of the CFU results is in accord with those of the TVBN results and the color-change phenomena of the smart tag, as shown in **Figure 2**. When the fresh shrimps have been stored at 25 °C for 1 day, the CFU value grows exponentially, so the freshness decreases sharply. When the shrimps have been stored at 4 °C, the CFU value decreases at first because of the death of the chill-intolerable microorganism, then significantly increases, which means the freshness begins to decrease. When the shrimps have been stored at -16 °C for 1-7 days, the CFU value also dramatically decreases at first, then increases slowly. The increase of bacteria means a decrease of the freshness. Comparing with the CFU value in the shrimps stored at 4 °C, the CFU value at -16 °C remains at a lower level. Thus, the shrimps are just slightly decomposed after 5 days. Therefore, our smart tags could be readily applied in the shrimp supply chain for real-time and visual monitoring the freshness of shrimps.

Figure 2. (a) Colony forming units (CFU) of bacteria in shrimps stored at 25 °C, 4 °C and -16 °C for different days; (b-d) The ratio of CFU and original CFU (fresh shrimps, 0 days) of bacteria in shrimps stored at 25 °C, 4 °C and -16 °C for different days.

(2) Reviewer 3 # wrote: *I am not satisfied with the set of other tested sea food, as there should be much more included. To my experience the shrimps is the worse choice to test because they are very easily spoiled and they have very strong initial odour. On the other hand, tuna is also a type of fish that is very sensitive in freshness; tuna could be not fresh any longer, but there is no amine-like odour released from it. Based on only three different seafoods and without validation and comparison with a reference method, authors cannot say about the real freshness of the seafoods and reliability of the results obtained. Authors also should define which seafood their method is appropriate for in order to 'readily provide practical applications of their sensors in food industry'.*

Answer: Thanks for your suggestion. We have revised the manuscript and narrowed our detection range to shrimp and crab, because of the obvious fluorescence color change of our smart tag.

Reviewers' Comments:

Reviewer #3:

Remarks to the Author:

The authors have revised the manuscript with respect to the issues raised. However, I still have remarks to be considered:

1. For which bacteria (name) do CFUs apply?
2. What is the CFU limit value (and for which bacteria/microorganisms) that determines the freshness of the shrimps? Please support this with an appropriate reference.
3. I do not agree with '... has provided more sensitive approach (compared to which method???) to access the fresh state of shrimps before totally spoilage', because it is not really possible to determine the freshness of the food using a newly developed probe without correlating its signal with microbiological activity (expressed in CFU). The number of bacteria that limits the freshness of the shrimps should be known and correlated with the signal of the sensor/probe (see Talanta 69, 2006, 515-520).
4. TVBN and CFU tests should be applied to crab too (if the crab is mentioned in the title) and commented similarly to results obtained with shrimps – it would be interesting to compare which of the tested seafood is more perishable in terms of freshness and how the smart tag responds in both cases in comparison with microbiological activity.
5. You should include in the introduction and in the conclusion part, that the method was validated against both methods, TVBN and microbial activity.
6. I recommend English proofing.

According to my opinion the paper could be accepted after taking into account the above issues.

Responses to the reviewers' comments and list of the details of the changes:

Answers to Comments by Reviewer # 3

(1) Reviewer 3 # wrote: *For which bacteria (name) do CFUs apply?*

Answer: Colony forming units was the total number of microbial colonies formed in the solid medium of bacteria in tested sample. The bacteria include *Escherichia coli*, *Enterococcus faecalis*, *Salmonella* and other contaminated bacteria in the food. The total bacterial counts are widely used as a conventional method to evaluate the hygienic standards of food and define the contamination degree of food by bacteria.

(2) Reviewer 3 # wrote: *What is the CFU limit value (and for which bacteria/microorganisms) that determines the freshness of the shrimps? Please support this with an appropriate reference.*

Answer: Actually, it's hard to determine the CFU limit value which represents the total spoilage of shrimps, because original CFU of shrimps is subjected to many factors, such as shrimp species, producing area and fishing time. There is no national standard about this issue. Generally, the change of CFU value during storage is used to evaluate the freshness of shrimps. A significant increase of CFU means the decrease of the shrimp freshness (*Food Control 2010, 21, 1263-1271* and *Journal of Food Science 2005, 70, S459-S466*). The relevant references have been added into the revised manuscript.

(3) Reviewer 3 # wrote: *I do not agree with '... has provided more sensitive approach (compared to which method???) to access the fresh state of shrimps before totally spoilage', because it is not really possible to determine the freshness of the food using a newly developed probe without correlating its signal with microbiological activity (expressed in CFU). The number of bacteria that limits the freshness of the shrimps should be known and correlated with the signal of the sensor/probe (see Talanta 69, 2006, 515-520).*

Answer: Thanks for your kind comments. We have deleted the sentence "Moreover,

our smart tag has provided a simple and more sensitive approach to assess the fresh state of shrimps before totally spoilage, i.e. slightly decomposed.” in the manuscript.

(4) Reviewer 3 # wrote: *TVBN and CFU tests should be applied to crab too (if the crab is mentioned in the title) and commented similarly to results obtained with shrimps - it would be interesting to compare which of the tested seafood is more perishable in terms of freshness and how the smart tag responds in both cases in comparison with microbiological activity.*

Answer: Thanks for your kind suggestion. The TVBN and CFU tests of the crab have been detected. The experiment results (Fig. S11 and S13) have been added in the revised manuscript. These experiment results are completely consistent with the color-change phenomena of the smart tag. Based on TVBN, CFU and smart-tag results, the shrimps are more perishable in terms of freshness, as shown in Fig. 4 and S9-13. For example, when the shrimps and crabs were stored at 25 °C for 1 day, the TVBN content of shrimps was about 132 mg/100g, while that of crabs was only about 15 mg/100g. The smart tag for monitoring shrimps changes its fluorescence color from red to green, while the smart tag for monitoring crabs is yellow. So, after storage for 1 day at 25 °C, the shrimps have been totally spoiling, while the crabs are not.

(5) Reviewer 3 # wrote: *You should include in the introduction and in the conclusion part, that the method was validated against both methods, TVBN and microbial activity.*

Answer: Thanks for your kind comments. We have added this content in the manuscript according to your suggestion.

(6) Reviewer 3 # wrote: *I recommend English proofing.*

Answer: Thanks for your kind comments. And, as your suggestion, our manuscript have been revised carefully by ACS Language Editors.

Reviewers' Comments:

Reviewer #3:

Remarks to the Author:

The authors have revised the manuscript with respect to the issues raised in the 3rd revision. Now, the content of the article is more supported by the analytical aspects in terms of freshness and could be published. The two last comments I have are:

1) About the explanation on the CFU limit value which corresponds to the freshness of the shrimps, which is the answer from previous revision. This explanation is somehow missing and should be integrated into the manuscript, along with the references.

(Explanation to be included:

'Actually, it's hard to determine the CFU limit value which represents the total spoilage of shrimps, because original CFU of shrimps is subjected to many factors, such as shrimp species, producing area and fishing time. There is no national standard about this issue. Generally, the change of CFU value during storage is used to evaluate the freshness of shrimps. A significant increase of CFU means the decrease of the shrimp freshness (Food Control 2010, 21, 1263-1271 and Journal of Food Science 2005, 70, S459-S466).')

2) Regarding the Figs. S10, S11, S12: in these Figs. the c) and d) diagrams have different y-scales (normal) than in the a) and b) (y-scale is logarithmic). Why is this so? The b), c) and d) should have similar y-scale, in logarithmic values, but they do not have. On the other hand, no such matter is observed on Fig. S13. Please, explain, correct if necessary.

Responses to the reviewers' comments and list of the details of the changes:

Answers to Comments by Reviewer # 3

(1) **Reviewer 3 # wrote:** *About the explanation on the CFU limit value which corresponds to the freshness of the shrimps, which is the answer from previous revision. This explanation is somehow missing and should be integrated into the manuscript, along with the references.*

Answer: Thanks for your kind suggestion. The explanation has been integrated into the manuscript.

(2) **Reviewer 3 # wrote:** *Regarding the Figs. S10, S11, S12: in these Figs. the c) and d) diagrams have different y-scales (normal) than in the a) and b) (y-scale is logarithmic). Why is this so? The b), c) and d) should have similar y-scale, in logarithmic values, but they do not have. On the other hand, no such matter is observed on Fig. S13. Please, explain, correct if necessary.*

Answer: Different scales were used to make sure that data were exhibited more clearly and intuitively. For example, the TVBN content increased in an exponential manner in Fig. S10-12 (b), while the TVBN content increased in the same order of magnitude in Fig. S10-12 (c) and (d). If the normal scale was used, some data in Fig. S10-12 (b) can't be clearly exhibited in the graph because of the huge difference between the maximum value and minimum value. If the logarithmic scale was used, some data in Fig. S10-12 (c) and (d) can't be clearly exhibited in the graph because of the small difference between the maximum value and minimum value. In Fig. S13, the TVBN content increased in an exponential manner in (b), (c) and (d).